# Androgen Affects the Inhibitory Avoidance Memory by Primarily Acting on Androgen Receptor in the Brain in Adolescent Male Rats

**DOI:** 10.3390/brainsci11020239

**Published:** 2021-02-14

**Authors:** Md Nabiul Islam, Yuya Sakimoto, Mir Rubayet Jahan, Emi Miyasato, Abu Md Mamun Tarif, Kanako Nozaki, Koh-hei Masumoto, Akie Yanai, Dai Mitsushima, Koh Shinoda

**Affiliations:** 1Division of Neuroanatomy, Department of Neuroscience, Yamaguchi University Graduate School of Medicine, 1-1-1 Minami-Kogushi, Ube 755-8505, Japan; nabiul@yamaguchi-u.ac.jp (M.N.I.); rubayet.lucky@gmail.com (M.R.J.); g096eb@yamaguchi-u.ac.jp (E.M.); mamuntarif45@gmail.com (A.M.M.T.); nozakik@yamaguchi-u.ac.jp (K.N.); masukh@yamaguchi-u.ac.jp (K.-h.M.); akiey@yamaguchi-u.ac.jp (A.Y.); 2Department of Physiology, Yamaguchi University Graduate School of Medicine, 1-1-1 Minami-Kogushi, Ube 755-8505, Japan; ysaki@yamaguchi-u.ac.jp (Y.S.); mitsu@yamaguchi-u.ac.jp (D.M.); 3Department of Anatomy and Histology, Bangladesh Agricultural University, Mymensingh 2202, Bangladesh; 4Department of Basic Laboratory Sciences, Faculty of Medicine and Health Sciences, Yamaguchi University Graduate School of Medicine, 1-1-1, Minami-Kogushi, Ube 755-8505, Japan

**Keywords:** emotional memory, male sex hormone, hippocampus, amygdala, rodent behavior

## Abstract

Adolescence is the critical postnatal stage for the action of androgen in multiple brain regions. Androgens can regulate the learning/memory functions in the brain. It is known that the inhibitory avoidance test can evaluate emotional memory and is believed to be dependent largely on the amygdala and hippocampus. However, the effects of androgen on inhibitory avoidance memory have never been reported in adolescent male rats. In the present study, the effects of androgen on inhibitory avoidance memory and on androgen receptor (AR)-immunoreactivity in the amygdala and hippocampus were studied using behavioral analysis, Western blotting and immunohistochemistry in sham-operated, orchiectomized, orchiectomized + testosterone or orchiectomized + dihydrotestosterone-administered male adolescent rats. Orchiectomized rats showed significantly reduced time spent in the illuminated box after 30 min (test 1) or 24 h (test 2) of electrical foot-shock (training) and reduced AR-immunoreactivity in amygdala/hippocampal cornu Ammonis (CA1) in comparison to those in sham-operated rats. Treatment of orchiectomized rats with either non-aromatizable dihydrotestosterone or aromatizable testosterone were successfully reinstated these effects. Application of flutamide (AR-antagonist) in intact adolescent rats exhibited identical changes to those in orchiectomized rats. These suggest that androgens enhance the inhibitory avoidance memory plausibly by binding with AR in the amygdala and hippocampus.

## 1. Introduction

In males, testosterone (T) is primarily produced in the testicles and it can be metabolized to dihydrotestosterone (DHT) by 5 α-reductase enzyme or can be aromatized to estrogen. Androgens can exert their effects either via binding with androgen receptor (AR) or by binding with estrogen receptors [1,2,3,4,5]. In addition, DHT can be further converted to 5α-androstane-3α, 17β-diol or 5α-androstane-3β, 17β-diol [6]. Furthermore, androgens may also exert their effects by membrane AR [7]. In the rodent brain, AR is highly present in the multiple brain areas, including bed nucleus of stria terminalis, preoptic area, hypothalamus, amygdala, and in the pyramidal neurons of cornu Ammonis 1 (CA1) subfield of hippocampus [8,9,10,11,12,13,14].

Adolescence is regarded as a critical neurodevelopmental stage for the action of androgen in the different regions of the brain [15]. Adolescence starts around postnatal day 28 in rodents [16]. Testosterone production in testis starts to augment in adolescent period of male rodents. Serum androgen concentration also begins to increase in this period of postnatal development [17,18,19]. AR is present in the diverse areas of the rodent brain and the expression of AR is increased duo to the effects of androgen around puberty [20,21]. Adolescence is the dynamic stage of postnatal neurodevelopment in which social and cognitive skills mature [22]. It is suggested that androgens may regulate the storage of learning and memory in the brain [23,24,25]. In particular, androgens affect spatial learning [26,27,28], novel object recognition memory [29], memory for visual objects [30], normal short-term memory [31], and anti-anxiety behavior [32,33]. 

Inhibitory avoidance task is an emotional conditioning paradigm in which the rodents learns to corelate a particular context with the occurrence of an aversive event (such as an electrical shock). During inhibitory avoidance training, rats usually receive a single aversive foot-shock after entering from an illuminated compartment into a darkened compartment. Retention of the training is tested later by measuring latency of rats to enter the former shock compartment when they are placed in the illuminated compartment. Longer retention test latencies are considered as better memory [34,35]. Effect of androgen on inhibitory avoidance has cursorily been reported in adult rats [36,37]. However, the effects of androgen on inhibitory avoidance memory have never been reported in adolescent rat. In the present study, we aimed to elucidate the effects of androgens (both aromatizable androgen, T and non-aromatizable androgen, DHT) on inhibitory avoidance memory in adolescent male rats. As the performance in the inhibitory avoidance task is largely dependent on the amygdala and the hippocampus [36,38], we also studied the changes of AR-expression in these brain areas after hormonal manipulations. Finally, to clarify the possible pathway of the action of androgens, we also set out to examine the effects of the flutamide (AR-antagonist) on the inhibitory avoidance memory and on AR-expression in the amygdala and hippocampus.

## 2. Materials and Methods

### 2.1. Rats and Ethical Statement 

Sprague-Dawley rats (3 weeks) were obtained from the Japan SLC, Inc. (Hamamatsu, Shizuoka, Japan) just after weaning. Rats were then transferred to the central Animal facility of the Yamaguchi University Graduate School of Medicine and maintained at a constant temperature (22–24 °C) and on a light/dark cycle (lights on 8AM to 8 PM) with sufficient food and water. All the research protocols of our current study were approved by the “Committee of Ethics on Animal Experiments of Yamaguchi University Graduate School of Medicine”. All the animal experiments of the present study were performed according to the “guidelines for Animal Experiments of Yamaguchi University Graduate School of Medicine” and the “Japanese Government (Notification no. 6, Law no. 105)”. Sufficient efforts were made to reduce the sufferings of rats during animal transfer, handling in different surgical experiments, or in postoperative recovery.

### 2.2. Surgical Procedure and Steroidal Manipulation

Rats were randomly divided into four experimental groups such as (1) sham-operated (sham-control), (2) orchiectomized (OCX), (3) OCX + T, and (4) OCX + DHT group (Figure 1A). All groups contained 18 rats. In the current study, all the surgical procedures were performed under general anesthesia with sodium pentobarbital (30 mg/kg, i.p.). For orchiectomy, the anesthetized rats were placed in supine position. After shaving the scrotal hair, at the tip of the scrotum a small median incision was made through the skin. Then, a single ligature was made on the spermatic cord after identifying testicular blood vessels, vas deferens, epididymis and testicle. The testis and epididymis were removed and the procedure were repeated for another testis. Finally, the cremaster muscle and scrotal skin were sutured. For the rats in the sham-operated group, the similar procedure was conducted only for visual identification of the testis and its associated structures and the scrotal skin was sutured without ligation or resection of testis. T or DHT capsules were placed subcutaneously in the neck on the same day of surgical operation. Immediately after surgical procedure, the rats were placed on an absorbent pad inside a heated recovery cage. The operated animals were regularly observed until they are ambulatory. After recovering from anesthesia, rats were moved to home cage. The day of surgical steroidal manipulation was counted as day 1 (postnatal day 29) and the inhibitory avoidance task were performed on day 7 (postnatal day 35). 

The silastic capsules for T and DHT were made according to our previous study [14]. First, powdered-T (Nacalai Tesque Inc., Kyoto, Japan) or powdered-DHT (Sigma-Aldrich, St. Louis, MO, USA) were inserted into a piece of silastic tube (outer diameter 3.0 mm, inner diameter 2.0 mm; 12 mm length per 100 g body weight; As One, Osaka, Japan). The silastic capsules were then kept in saline at 4°C until use. 

To examine the influence of the flutamide (F9397, Sigma-Aldrich) on the AR-immunoreactivity and inhibitory avoidance memory, rats were assigned randomly to other two experimental groups: (1) intact + vehicle (sesame oil, S3547, Sigma-Aldrich) (*n* = 18), (2) intact + flutamide treated group (*n* = 18). In the current study, flutamide was injected subcutaneously (60 mg/kg) once daily for 4 days (postnatal day 32, 33, 34 and 35) and the inhibitory avoidance task were performed after 2–3 h of the last injection [14]. All efforts were made to calm the rats prior to injections to lessen any stress. Before injection, rats were held with one hand and swung gently from side to side to make them calm. The volume of injection was also kept same for each administration [14,39]. 

### 2.3. Inhibitory Avoidance Task

The inhibitory avoidance task apparatus (width: 58 cm, length: 33 cm, height: 33 cm) was a 2-chambered box consisting of an illuminated safe side and a dark electric shock side separated by a trap door [40,41,42]. For training, adolescent rats were placed in the illuminated side of the box facing a corner opposite the door. After the trap door was opened, the rats could enter the dark box at their will. The time spent before entering the novel dark chamber was measured as a behavioral parameter. Soon after the rats entered the dark side, the door was closed and a scrambled electrical foot-shock (1.6 mA, 2 s) was applied with electrified steel rods in the floor of the dark compartment (inhibitory avoidance training). The rats were kept in the dark box for 10 s before being returned to their home cage. Thirty minutes after the procedure described above, the rats were placed in the light side (inhibitory avoidance test 1). Twenty-four hours after the foot-shock, the rats were placed in the light side again (inhibitory avoidance test 2). The time spent in the illuminated chamber before entering the dark one was measured as an indicator of learning performance.

### 2.4. Serum Steroid Hormone Assay

For serum steroid immunoassay, blood was collected from the ventricle of the heart of anesthetized rats using 23 G needle with 5 mL syringe. Serum T and DHT concentrations were measured with enzyme-linked immunosorbent assay (ELISA) kit (T, RE52151; DHT, DB52021; Immuno-Biological Laboratories, Hamburg, Germany) following our previous studies [11,14]. Each sample was assayed in duplicate in a single ELISA.

### 2.5. Primary Antibodies

Primary antibody details used are described in Table 1. All the primary antibodies were selected from the Antibody Registry of the Neuroscience Information Framework. All the antibodies were commercially purchased. The specificity of all the primary antibodies used in current study was determined in our previous studies (Table 1).

### 2.6. Western Blotting

The Western blotting was performed according to our earlier studies [44,45]. The dissected amygdaloid and hippocampal tissues were homogenized with T-PER tissue protein extraction reagent (Thermo Scientific, Chicago, IL, USA) mixed with 5 μL/mL of protease inhibitor (P8340; Sigma-Aldrich). Proteins samples were separated with 7.5% SDS-PAGE gels electrophoresis and then transferred onto polyvinylidene fluoride (PVDF) membrane by wet transfer in 4 °C for 2 h. The PVDF membranes were blocked with 5% skimmed milk for 60 min at 20 °C and then probed overnight at 4 °C with rabbit polyclonal anti-AR antibody (Abcam) or mouse monoclonal anti-α tubulin antibody (Sigma-Aldrich). After washing three times, the membranes were incubated with specific secondary antibodies (horseradish peroxidase-linked anti-rabbit or anti-mouse IgG; GE Healthcare, Amersham, Buckinghamshire, UK; 1:20,000 dilution) at 20 °C for 2 h. After several washes, the immunopositive reactions were visualized with ECL select (GE Healthcare) solutions and captured using an Amersham 600 Imager (GE Healthcare). Finally, the NIH ImageJ software (version for Windows, Bethesda, MD, USA) was used to semi quantitative analyses of the immunoreactive bands.

### 2.7. Immunohistochemistry

Adolescent male rats were transcardially perfused with 4% paraformaldehyde solution (pH 7.4). After removing from the skull, the brains were post-fixed for overnight at 4 °C in the same perfusion solution and then soaked in 30% sucrose solution at 4 °C for 5–7 days. Frontal brain sections were made with a cryostat at a thickness of 30 μm. The coordinates of hippocampus and amygdala were 2.80 to 3.30 mm posterior to the bregma (according to “The rat brain in stereotaxic coordinates” by George Paxinos and Charles Watson, fourth edition). 

Details for the methodology of immunohistochemistry were described in our earlier studies [43,46]. In brief, pre-incubation of the sections was done with PBS containing 10% normal goat serum (NGS) and 0.3% Triton X-100 at 4 °C for 60 min. A mixture of 50% methanol and 1.5% hydrogen peroxide for 30 min at 4 °C were used for bleaching of sections. The sections were then incubated with anti-AR primary antibody (Santa Cruz Biotechnology, Santa Cruz, CA, USA) for 3 days at 20 °C. After washing three times, the sections were incubated with biotinylated goat anti-rabbit secondary antibody (Dako, Glostrup, Denmark; 1:1000 dilution) for overnight at 4 °C. After washing three times, the sections were then incubated with peroxidase-conjugated streptavidin (Dako, Glostrup, Denmark; 1:1000 dilution) at 20 °C for 3 h. For development of positive reactions, the brain sections were then incubated with diaminobenzidine-nickel solution for 20–30 min on ice. The sections were then mounted on gelatin-coated glass slides. After air-dry in room temperature for 30 min, the sections were dehydrated using an ascending graded series of ethanol, dipped twice in xylenes, and embedded using Entellan New mounting medium (107961; Merck Millipore, Billerica, MA, USA). A color digital camera (Lumenera Corporation, Ottawa, ON, Canada) equipped with a Nicon Eclipse 80i photomicroscope (Tokyo, Japan) was used for taking the photomicrograph of AR-immunoreactivity. 

### 2.8. Statistical Analysis

The time spent in illuminated box (inhibitory avoidance task), serum steroid levels, and AR-immunoreactivity in the amygdala or hippocampus among sham-control and steroid-manipulated groups were analyzed by one-way ANOVA followed by Scheffe’s post hoc tests. However, between the adolescent intact and AR-antagonist treated rats the above-mentioned parameters were analyzed by student’s *t*-test. In each test, *p* values of <0.05 were set as statistically significant. All the statistical analyses were performed using an SPSS software (version 22 for Windows; SPSS Inc., Chicago, IL, USA).

## 3. Results

### 3.1. Serum Steroid Levels after Manipulations of T and DHT

The serum concentrations of both T (Sham, 1.71 ± 0.25 ng/mL; OCX, 0.29 ± 0.09 ng/mL; *F*
_(3,68)_ = 312.75; one-way ANOVA followed by Scheffe’s post hoc test, *p* < 0.001) and DHT (Sham, 171.35 ± 12.29 pg/mL; OCX, 41.03 ± 8.27 pg/mL; *F*
_(3,68)_ = 369.99; one-way ANOVA followed by Scheffe’s post hoc test, *p* < 0.001) were significantly decreased in OCX rats in comparison to sham-operated rats (Table 2). The administration of T in OCX rats recovered the serum concentration of both T (2.21 ± 0.27 ng/mL) and DHT (221.35 ± 31.13 pg/mL) to normal levels; whereas, administration of DHT (Table 2) did not recover serum T level (0.41 ± 0.15 ng/mL). DHT only recovered the serum DHT level (233.62 ± 28.21 pg/mL) to normal level (Table 2). 

### 3.2. Effects of Gonadal Steroids on Inhibitory Avoidance Memory in Adolescent Male Rats

To examine the effects of androgen on emotional memory, adolescent male rats were examined for inhibitory avoidance task. In this emotional learning paradigm, adolescent rats were allowed to cross from an illuminated box to a dark one, where an electric foot-shock (2 s, 1.6 mA) was applied (Figure 1B). In training, rats from sham-operated or hormonal manipulated groups showed almost similar shorter time spent in the illuminated box before entering to dark (Figure 1C; Sham, 23.51 ± 2.79 s; OCX, 23.83 ± 3.19 s; OCX + T, 23.11 ± 4.05 s; OCX + DHT, 24.33 ± 3.25 s; *F*
_(3,23)_ = 0.02; one-way ANOVA followed by Scheffe’s post hoc test, *p* = 0.98). Then, we measured the time spent in the illuminated box thirty minutes (Test 1) and 24 hours (Test 2) after the inhibitory avoidance task. In test 1, the time spent in the illuminated box was significantly decreased in OCX rats in comparison to sham-operated rats. Administration of either T or DHT to OCX rats induced a significant increase in time spent in the illuminated box in test 1 (Figure 1D; Sham, 301.66 ± 22.79 s; OCX, 143.50 ± 8.18 s; OCX + T, 309.83 ± 12.05 s; OCX + DHT, 283.83 ± 3.25 s; *F*
_(3,23)_ = 9.79; one-way ANOVA followed by Scheffe’s post hoc test, *p* < 0.001). Similarly, in test 2, the time spent was also significantly decreased in OCX rats in comparison to sham-operated animals and administration of T or DHT to OCX rats resulted in a significant increase in time spent in the illuminated box (Figure 1E; Sham, 308.32 ± 30.71 s; OCX, 126.83 ± 9.21 s; OCX + T, 307.84 ± 23.02 s; OCX + DHT, 262.96 ± 19.17 s; *F*
_(3,23)_ = 18.15; one-way ANOVA followed by Scheffe’s post hoc test, *p* < 0.001). In our current study, both tests (short time, 30 min after training and long time, 24 h after training) showed similar effects of androgens on the duration of time spent the in the illuminated box after electrical foot shock. These results suggested that androgen modulated both the short-term and long-term inhibitory avoidance memory in adolescent rats. 

### 3.3. Effects of Gonadal Steroids on the Expression of AR in the Amygdala and Hippocampus in Adolescent Male Rats 

In our previous study, we showed the effects of androgen on AR-immunoreactivity in hippocampus in adolescent rats [14]. However, as both the amygdala and hippocampus have important roles in mediating the inhibitory avoidance memory, here we examined the expression of AR in these brain areas after hormonal manipulations and compared with sham-operated rats employing Western blotting and immunohistochemistry. In Western blotting, the OCX rats exhibited a significant depletion in AR-immunoreactivity both in amygdala and in the hippocampus in comparison to sham-controls. When T or DHT was applied to OCX rats, either T or DHT supplementation resulted in a significant enhancement of AR-immunoreactivity both in the amygdala (Figure 2A,B; Sham, 59.65 ± 7.92%; OCX, 15.91 ± 6.23%; OCX+ T, 60.33 ± 6.15%; OCX+ DHT, 61.73 ± 6.17%; *F*
_(3,20)_ = 62.27; one-way ANOVA followed by Scheffe’s post hoc test, *p* < 0.001) and in hippocampus (Figure 2A,C; Sham, 57.98 ± 8.75%; OCX, 14.25 ± 5.82%; OCX+ T, 61.00 ± 8.02%; OCX+ DHT, 66.75 ± 6.39%; *F*
_(3,20)_ = 63.01; one-way ANOVA followed by Scheffe’s post hoc test, *p* < 0.001).

In immunohistochemistry, high amount of AR-immunoreactivity was present in the nuclei of the medial amygdala (Figure 2D) and in the CA1 pyramidal neurons (Figure 2E) in sham-operated rats. Compared with control rats, medial amygdaloid or pyramidal neurons of CA1 subfield in OCX rats showed a sharp depletion in AR-immunoreactivity, and the immunoreactions seemed to translocate to the cytoplasm from the nucleus (Figure 2F,G). The application of either T (Figure 2H,I) or DHT (Figure 2J,K) in OCX male rats caused in the augmentation of AR-immunoreactivity in those brain areas. Both DHT and T application reversed the cytoplasm to nuclear translocations of the AR-immunoreactivity in the amygdaloid and in the CA1 pyramidal neurons (Figure 2H–K). The immunohistochemical and Western blotting results of hippocampus were consistent with our previous study [14].

The aforementioned inhibitory avoidance task and immunohistochemical data showed that both non-aromatizable DHT and aromatizable T exerted similar effects on the restoration of AR-immunoreactivity in the amygdala and CA1 hippocampal subfield as well as on the improvement of time spent in the illuminated box after 30 min or 24 hours of having foot shock. These suggest that androgen might modulate inhibitory avoidance memory by primarily binding with AR. To clarify whether androgens affect AR-immunoreactivity and inhibitory avoidance memory by binding with AR, intact adolescent male rats were treated with the AR-antagonist flutamide.

### 3.4. Effects of Flutamide on Serum Androgen Concentration and Inhibitory Avoidance Memory in Adolescent Male Rats 

In the present study, intact adolescent male rats were treated with flutamide for four consecutive days (Figure 3A). Daily flutamide injection resulted significant augmentation of serum T (Figure 3B; vehicle-treated, 1.78 ± 0.18 ng/mL; flutamide-treated, 2.50 ± 0.21 ng/mL; *t*
_(34)_ = −7.79; Student’s *t*-test, *p*  <  0.001) or serum DHT levels (Figure 3B; vehicle-treated, 180.33 ± 8.18 pg/mL; flutamide-treated, 272.83 ± 15.21 pg/mL; *t*
_(34)_ = −12.56; Student’s *t*-test, *p* < 0.001) compared with those in intact vehicle-treated adolescent rats.

Next, we examined the effects of AR-antagonist on the inhibitory avoidance memory (Figure 3C). In training, rats from intact vehicle-treated or flutamide-treated groups showed similar shorter latency period in the illuminated box (Figure 3D; intact vehicle-treated, 25.84 ± 2.79 s; intact flutamide-treated, 28.39 ± 4.62 s; *t*
_(10)_ = −0.20; *p*  =  0.85; Student’s *t*-test). Then, we measured the time spent in the illuminated box 30 minutes (Test 1) and 24 hours (Test 2) after the inhibitory avoidance task. In test 1, the time spent in the illuminated box was significantly reduced in flutamide-treated rats in comparison to intact vehicle-treated rats (Figure 3E; vehicle-treated, 308.17 ± 19.18 s; flutamide-treated, 133.17 ± 5.22 s; *t*
_(10)_ = 6.41; Student’s *t*-test, *p* <  0.001). Similarly, in test 2, the time spent was also significantly reduced in flutamide-treated rats in comparison to intact vehicle-treated rats (Figure 3F; vehicle-treated, 304.83 ± 21.94 s; flutamide-treated, 127.16 ± 12.39 s; *t*
_(10)_ = 7.04; Student’s *t*-test, *p* <  0.001).

### 3.5. Effects of Flutamide on AR-immunoreactivity in the Amygdala and Hippocampus 

In Western blotting, in comparison to intact adolescent rats, the AR-antagonist-treated rats showed a significant depletion in AR-immunoreactivity both in the amygdala (Figure 4A,B; vehicle-treated, 67.30 ± 4.18%; flutamide-treated, 24.95 ± ± 12.39 s; *t*
_(10)_ = 7.25; Student’s *t*-test, *p* <  0.001) and in the hippocampus (Figure 4A,B; vehicle-treated, 63.46 ± 3.71%; flutamide-treated, 29.63 ± 3.83 s; *t*
_(10)_ =6.82; Student’s *t*-test, *p* <  0.001). 

Similarly, in immunohistochemistry, the both amygdaloid (Figure 4C,E) and pyramidal CA1 neurons (Figure 4D,F) of flutamide-injected rats exhibited a sharp reduction in AR-immunoreactivity compared with intact adolescent rats.

The aforementioned behavioral and immunohistochemical results clearly indicated that the orchiectomy and flutamide-treatment had similar effects on inhibitory avoidance memory as well as on the AR expression in the amygdala and hippocampus. Our results suggested that androgen might enhance inhibitory avoidance memory by primarily acting on AR in the amygdala and hippocampus.

## 4. Discussion

The current study, using rodent behavior technique, Western blot, and immunohistochemistry may provide the first detailed evidence that androgen affects the inhibitory avoidance memory in male adolescent rats. We observed that OCX rats exhibited significant decrease in AR-immunoreactivity in amygdala and in CA1 hippocampal subfield, and the time spent in the illuminated box after 30 min or 24 hours of having electrical foot-shock in comparison to sham-operated rats. Treatment of OCX rats with either non-aromatizable DHT or aromatizable T successfully recovered these effects. These suggest that androgens affect the inhibitory avoidance memory plausibly by binding with AR in the amygdala and hippocampus.

Inhibitory avoidance is an emotional conditioning paradigm in which the rats learn to associate a specific context with the occurrence of an aversive event [34,35]. Our present study is the first to provide clear evidence that androgens regulate short-term and long-term inhibitory avoidance memory in adolescent rats. Previous evidence suggests that orchiectomy decreases cognitive performance in T-maze, object recognition, and inhibitory avoidance memory in adult rats [23,33,36,47,48]. Androgen replacement to adult OCX rats enhances these cognitive performances. Furthermore, testosterone injection to aged male mice improves cognition in a foot-shock avoidance task [49]. It has been reported that androgen decline in the aging male (ADAM) causes significant memory loss [50,51] and testosterone replacement ADAM patient can improve their cognitive function [52,53,54]. In addition, older aged men administered with testosterone shows improved working and spatial memory [55,56]. Cognitive performance is affected in another condition of men, hypogonadism, where androgen declines independent of aging [57]. Testosterone injection to hypogonadal young adult improves their cognitive function [58]. In our current study, we observed that OCX rats showed significantly decreased latency period in the illuminated box after electric foot-shock and these effects were successfully reversed by the treatment of OCX rats with either non-aromatizable DHT or aromatizable T. The findings of our current study confirm and provide evidence that androgens can also affect emotional behavior in the adolescence stage of postnatal neurodevelopment where social and cognitive skills mature.

The amygdala and hippocampus play important roles in regulating the inhibitory avoidance memory [36,38]. In our current study, we examined the expression of AR in these brain areas in young adolescent male rats. Employing immunohistochemistry and Western blotting, we detected prominent AR-immunoreactivity both in the amygdala and in the hippocampus. These findings are consistent with earlier works [14,59,60,61]. In both immunohistochemistry and Western blotting, we detected that decreased AR-immunoreactivity in the amygdala and hippocampus of OCX rats, whereas AR-immunoreactivity were augmented in either DHT- or T-treated OCX rats. The effects of androgen on AR-immunoreactivity in the amygdala or in hippocampus of male rats in our present study is similar to that reported previously in the in different brain regions including the hippocampus, preoptic area, hypothalamic and amygdaloid areas of young or adult rodents [14,20,62]. It has been known that among the amygdaloid nuclei the basolateral amygdala is primarily responsible for the association between the conditioned stimulus (dark compartment) and unconditional stimulus (foot shock) [63]. In the present study, we observed prominent AR-immunoreactivity in the medial amygdala. Our results are in agreement with a previous work [64], where they also observed stronger AR-immunoreaction in the medial and central amygdala, but very lower AR-expression in the basolateral amygdala. However, it has been reported that there is a strong interconnection among the amygdaloid nuclei. The lateral nuclei receive moderate to heavy inputs from and project outputs to the central or medial nucleus [65]. In addition, inputs from the hippocampal area to the lateral nucleus of the amygdala originate in the entorhinal cortex, and subiculum and in the CA1 [66]. On the other hand, the lateral nuclei of the amygdala also provide direct input to the hippocampal formation [65]. Furthermore, the lateral amygdaloid nuclei receive substantial inputs form the hypothalamus [65] where AR is highly expressed [20,62]. It has been claimed that hypothalamus can act as a primary coordinator of memory updating [67]. Although it is not clearly understood how hypothalamus can participate in androgen-related memory functions, it is tempting to speculate that hypothalamus may act as a hub.

Evidence suggests that androgens can enhance neuronal excitability in the brain to modulate cognitive functions. In rodents and primates, orchiectomy decreases the density of synapses of hippocampal CA1 pyramidal neurons. These effects of androgen are reversed by androgen administration [37,68,69,70]. In addition to synaptic plasticity, in our previous study, we showed that androgen can also affect the intrinsic excitability of CA1 hippocampal pyramidal neurons in adolescent male rats [14]. Interestingly, some previous findings implicate that inhibitory avoidance task elevates firing rates in the central and basolateral amygdaloid neurons [71]. Although further detailed electrophysiological experiment need to be done to elucidate in what mechanism androgens regulate amygdaloid/hippocampal CA1 neuron to enhance inhibitory avoidance memory, taking in account the previous results, our current findings may suggest that androgen increase the basal sensitivity and excitability of amygdaloid/CA1 pyramidal neurons, that may influence selective activation of neuronal circuit or excitation to affect emotional memory of adolescent male rats. 

In the current study, we also studied the possible pathway of the action of androgens on the inhibitory avoidance memory. Our present data exhibit interesting evidence that both non aromatizable DHT and aromatizable T have identical effects on AR-immunoreactivity and on the latency in inhibitory avoidance task. These suggest that androgens mediate their effects directly by binding with AR not indirectly by binding with estrogen receptor. To clarify this hypothesis, we injected flutamide to intact adolescent male rats. In the present study, flutamide treatment resulted in augmentation of serum androgen levels. Flutamide blocks androgen-mediated negative feedback of pituitary luteinizing hormone release and, therefore, one would anticipate a sharp rise in serum androgen levels after flutamide treatment [14,72]. However, despite higher serum androgen concentration, flutamide sharply declines AR-immunoreactivity in the amygdala and in the hippocampus. The effects of flutamide on AR-immunoreactivity in the amygdala and hippocampus of male adolescent rats in our present study are consistent with previous studies [14,73]. However, by what mechanism flutamide downregulates AR-immunoreactivity has not yet been clearly known. It has been reported that flutamide can compete with androgens for AR binding but do not upregulate AR [74]. It has also been proposed that AR-antagonist might prevent the stable DNA binding of the AR to obstruct the functions of AR [75]. In addition, our current study also clarifies that intact flutamide-treated adolescent rats show significantly reduced latency period in the illuminated box after electric foot-shock compared with vehicle-treated intact rats. These changes are identical to those observed in OCX rats, indicating that androgens may affect inhibitory avoidance memory by primarily acting on AR in amygdala and hippocampus. However, it is possible that alternative mechanisms can also be involved. It is well known that DHT can also be metabolized to 5α-androstan-3β, 17 β-diol or to 5α-androstan-3α, 17 β-diol and may exert its effect via estrogen receptor β or GABA-A receptors respectively [6,57,76,77]. Furthermore, androgens also can act via membrane AR [78]. Future detailed studies are needed to reveal the exact pathway of the action of androgens on the inhibitory avoidance memory. In addition, it is also an important challenge to elucidate sex differences in the effects of androgen on the inhibitory avoidance memory, as there are prominent sex differences in AR-immunoreactivity in the different regions of brain [11]. 

## 5. Conclusions

Taken together, our current study is the first to elucidate that androgen can enhance the inhibitory avoidance memory in male adolescent rats after 30 min or after 24 hours of inhibitory avoidance training. Our current results confirm that the effects of T are reproduced by DHT and are not replicated by the application of flutamide. These indicate that the effects of androgens on inhibitory avoidance memory might primarily be mediated by androgen-specific receptor system particularly in amygdala and in hippocampus. Additional studies are required to clarify the exact mechanism of androgenic modulation of inhibitory avoidance memory.

## Figures and Tables

**Figure 1 brainsci-11-00239-f001:**
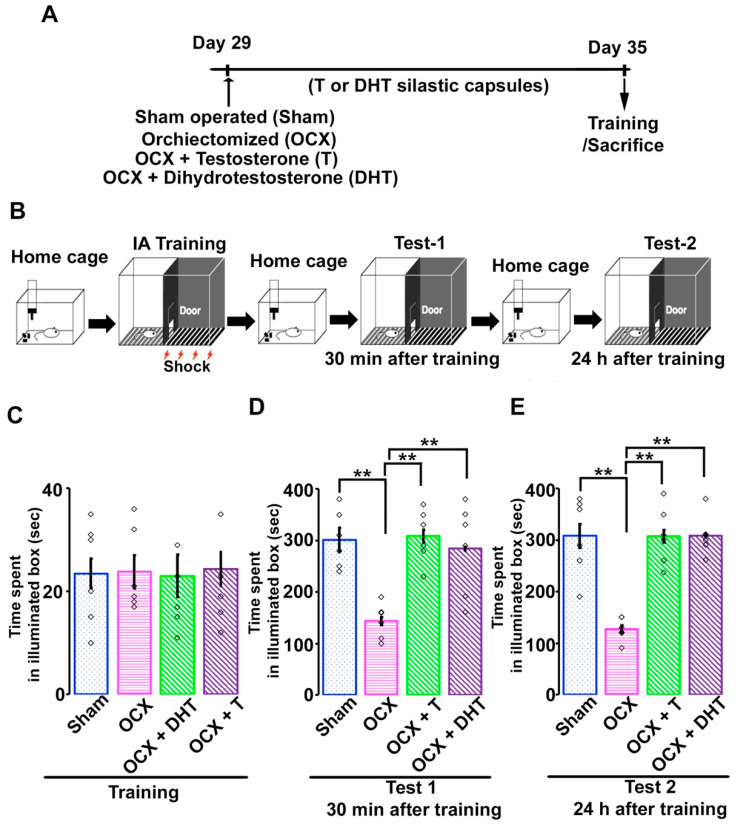
Time spent in illuminated box before entering the dark compartment of the box in sham-operated and steroid-manipulated male rats. (**A**) Schematic diagram of the schedule for surgical operations. (**B**) Diagram of experimental design and inhibitory avoidance task. A brief electrical foot-shock (2 s) was applied in the dark shock cage. (**C**) Time spent in illuminated box before training and (**D**) thirty minutes or (**E**) 24 hours after training of sham-operated and hormonal-manipulated rats. Values are the mean ± SEM. ***p* < 0.001. *n* = 6 for each group. White rectangles in (**C**–**E**) represent data points.

**Figure 2 brainsci-11-00239-f002:**
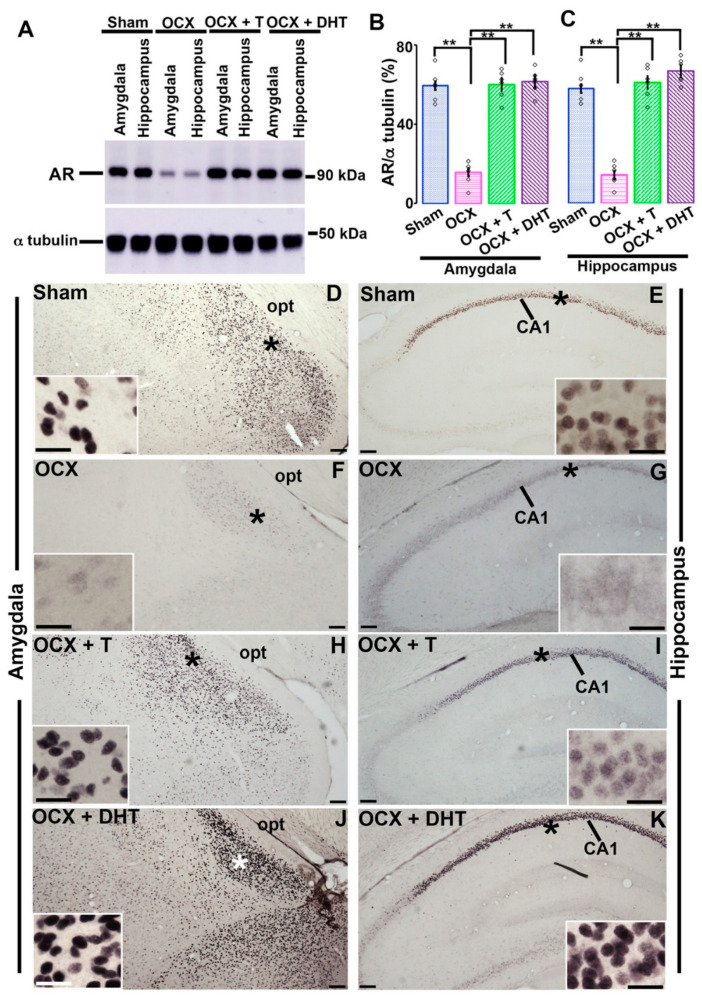
Androgen receptor (AR)-immunoreactivity in the amygdala and hippocampus of the sham-operated and steroid-manipulated male rats. (**A**) AR-immunoreactivity in the amygdaloid and hippocampal area were analyzed by Western blotting. (**B**,**C**) Total AR-immunoreactivity was normalized to α-tubulin. Values are the mean ± SEM. ***p* < 0.001. *n* = 6 for each group. White rectangles in (**B**,**C**) represent data points. (**D**–**K**) Photomicrographs showing the AR expression in the amygdala (**D**,**F**,**H**,**J**) and hippocampus (**E**,**G**,**I**,**K**) of steroid manipulated rats. Insets are the enlargements of * in (**D**–**K**), respectively. Scale bar = 100 μm in (**D**–**K**) and 20 μm in insets. opt, optic tract.

**Figure 3 brainsci-11-00239-f003:**
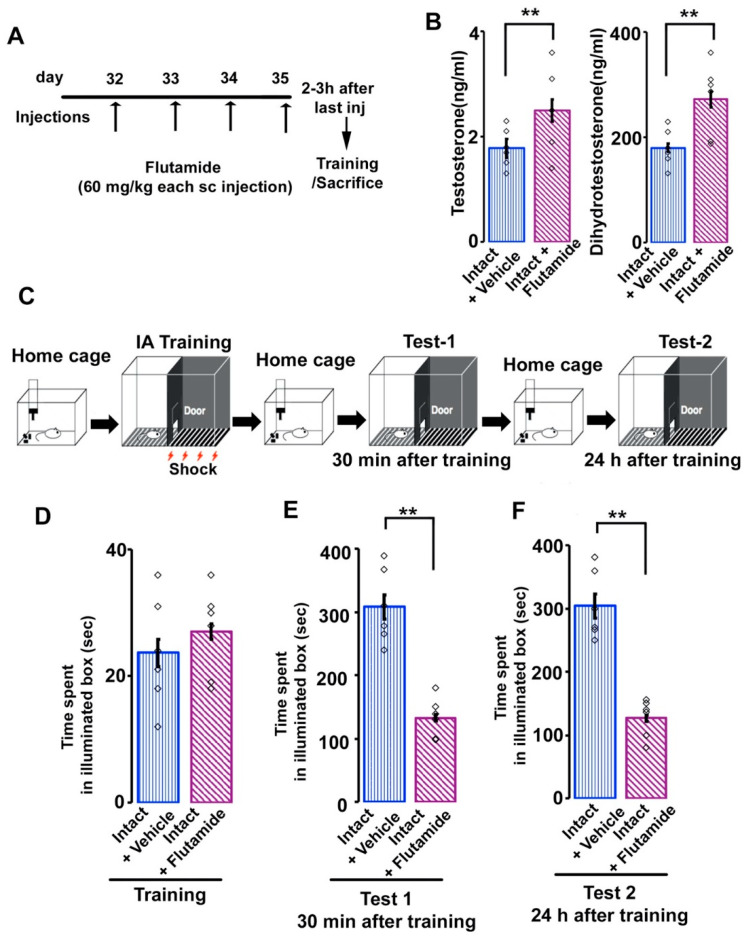
Time spent in illuminated box before entering the dark side of the box in vehicle and flutamide-injected male rats. (**A**) Schematic diagram of the schedule for flutamide injections. (**B**) Serum testosterone (T) and dihydrotestosterone (DHT) concentra-tion. *n* = 18 for each group. (**C**) Diagram of experimental design of inhibitory avoidance task. Rats were housed in a home cage but moved into the light box used for the task on the training day. A brief electrical foot-shock (2 s) was applied in the dark box in the shock cage. (**D**) time spent in illuminated box before training, (**E**) 30 minutes after training or (**F**) 24 h after training of vehicle and flutamide-treated intact rats. Values are the mean ± SEM. ***p* < 0.001. *n* = 6 for each group. White rectangles in (**B**), (**D**–**F**) represent data points.

**Figure 4 brainsci-11-00239-f004:**
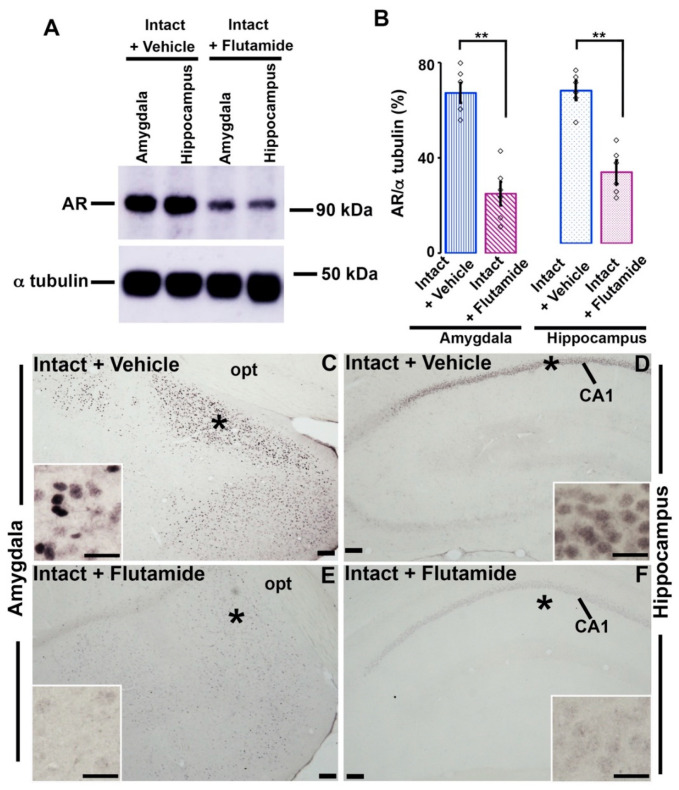
AR-expression in the amygdala or hippocampus in the intact and flutamide-injected male rats. (**A**) AR-immunoreactivity in the brain tissues containing the amygdaloid or hippocampal area were analyzed by Western blotting. (**B**) Total AR was normalized to α-tubulin. *n* = 6 for each group. Values are the mean ± SEM. ***p* < 0.001. Photomicrographs showing AR-immunoreactivity in (**C**,**E**) amygdala and (**D**,**F**) CA1 hippocampus of vehicle and AR antagonist-treated intact rats. Insets are the enlargements of * in (**C**–**F**) respectively. Scale bar = 100 μm in (**C**–**F**) and 20 μm in insets. opt, optic tract. White rectangles in (**B**) represent data points.

**Table 1 brainsci-11-00239-t001:** Primary antibodies used in the present study.

Name	Immunogen	Clonality/Host	Source	Dilution	References
Androgen receptor	Mouse androgen receptor	Polyclonal Rabbit	RRID: AB_1563391, Cat# sc-816, Santa Cruz Biotechnology, Santa Cruz, CA, USA	1:2000	[11,14]
Androgen receptor	Androgen Receptor from Human (N terminal, aa 1–10)	Monoclonal Rabbit	RRID: AB_11156085, Cat# ab133273, Abcam, Cambridge, UK	1:5000	[14]
α-tubulin	chicken embryonic brain derived microtubule	Monoclonal Mouse	RRID: AB_477583,Cat# T6199, Sigma-Aldrich, St. Louis, MO, USA	1:200,000	[43]

RRID, research resource identifier.

**Table 2 brainsci-11-00239-t002:** Serum androgen concentrations of adolescent rats after orchiectomy (OCX), treatment with testosterone (T), dihydrotestosterone (DHT). Values represented as the mean ± SEM.

Group	Number of Rats	T (ng/mL)	DHT (pg/mL)
Sham-operated	18	1.71 ± 0.25	171.35 ± 12.29
OCX	18	0.29 ± 0.09 ^a^	41.03 ± 8.27 ^a^
OCX + T	18	2.21 ± 0.27 ^b^	221.35 ± 31.13 ^b^
OCX + DHT	18	0.41 ± 0.15 ^a^	233.62 ± 28.21 ^b^

^a^*p* < 0.001 vs. Sham-operated controls; ^b^
*p* < 0.001 vs. OCX.

## Data Availability

All data generated or analyzed during this study are included in this published article.

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
