# Peer review of "Androgen Affects the Inhibitory Avoidance Memory by Primarily Acting on Androgen Receptor in the Brain in Adolescent Male Rats"

_brainsci, 2021, doi:10.3390/brainsci11020239_

Round 1
Reviewer 1 Report
After reviewing the manuscript and reading the rebuttal letter, I don't have any other comments.
The paper remains not original.
Author Response
After reviewing the manuscript and reading the rebuttal letter, I don't have any other comments. The paper remains not original.
Reply: We would like to thank the reviewer for his/her effort in reviewing this work. As the reviewer has no further comments, we attached the updated manuscript with the changes requested by another reviewer and academic editor. For the originality issue, we changed the text that was overlapped with that of our previous publications.
Reviewer 2 Report
The paper entitled " Androgen affects the inhibitory avoidance memory by primarily acting
on androgen receptor in the brain in adolescent male rats", which was resubmitted for
consideration by Islam et al. has been strengthened by the responses to previous criticisms.
The paper presents an extensive and compelling set of experiments demonstrating an
apparent androgen-receptor driven modulation of emotional memory in adolescent male rat.
The methodology is appropriate, the experiments are well documented, and the outcomes are
clear. These are novel observations in terms of an apparent selective AR modulation in males
of adolescent age group.
My suggestions for improvements are minor.
1.The author provided a schematic diagram of the schedule for flutamide injection (Fig.
3A). Similarly, the authors can add a schematic diagram for the schedule of hormonal
manipulations as a part of figure 1, which will be helpful for the potential readers.
2. First letter of each keywords (below the abstract) should be written in lowercase.
3. In materials and methods, the authors mentioned that rats were maintained in light/dark cycle. Which kind of light-dark cycle rat were subjected to? Please clarify.
4. Specify the catalog number/RRID of serum hormone ELISA kit.
5. If the author has already supplied the graphical abstract as I commented previously, then it's fine.
Author Response
The paper entitled "Androgen affects the inhibitory avoidance memory by primarily acting on androgen receptor in the brain in adolescent male rats", which was resubmitted for consideration by Islam et al. has been strengthened by the responses to previous criticisms. The paper presents an extensive and compelling set of experiments demonstrating an apparent androgen-receptor driven modulation of emotional memory in adolescent male rat. The methodology is appropriate, the experiments are well documented, and the outcomes are clear. These are novel observations in terms of an apparent selective AR modulation in males of adolescent age group. My suggestions for improvements are minor.
Reply: We would like to thank to the reviewer for giving his/her time and effort to review this manuscript.
- The author provided a schematic diagram of the schedule for flutamide injection (Fig.3A). Similarly, the authors can add a schematic diagram for the schedule of hormonal manipulations as a part of figure 1, which will be helpful for the potential readers.
Reply: We added the schematic diagram for the schedule of hormonal manipulations as part of figure 1 (Fig. 1A of revised version). We have also revised the figure legend accordingly.
- First letter of each keywords (below the abstract) should be written in lowercase.
Reply: We changed the keywords accordingly (indicated in red).
- In materials and methods, the authors mentioned that rats were maintained in light/dark cycle. Which kind of light-dark cycle rat were subjected to? Please clarify.
Reply: Following the suggestion of the reviewer, we clarified the light-dark circle (light on 8AM and off at 8PM, indicated in red, Materials and methods, page 2).
4. Specify the catalog number/RRID of serum hormone ELISA kit.
Reply: We specified the catalog number of serum hormone ELISA kit. (indicated in red, Materials and methods, page 4).
- If the author has already supplied the graphical abstract as I commented previously, then it's fine.
Reply: We submitted the graphical abstract.
We hope that this revised manuscript will address all of your concerns. Thank you very much for considering this revised version for publication.
This manuscript is a resubmission of an earlier submission. The following is a list of the peer review reports and author responses from that submission.
Round 1
Reviewer 1 Report
The manuscript by Md Nabiul Islam et al. is aimed at evaluating the effects of orchiectomy on the expression of androgen receptors in the amygdala and hippocampus, and on inhibitory avoidance memory in adolescent male rats. The findings are expected and, not surprisingly, are rescued by testosterone and dihydrotestosterone administrations. In fact, the Authors reported previous literature totally supported the interaction between androgens and cognitive functions, especially related to learning and memory. It is also known the role played by androgens during the adolescence, by its nature a critical period in which hormonal levels sustain the cognitive functions and decrease anxiety level, guaranteeing thus the exploration and the survival. On such a basis, the rationale would be more aimed and persuasive.
I have also other critics that can be useful to the Authors.
The Authors should accurately report the right References to support what they said. Only as an example, the reference number 26 does not seem associated to androgen/spatial learning, but to MAPK/spatial memory (Blum, S.; Moore, A.N.; Adams, F.; Dash, P.K. A mitogen-activated protein kinase cascade in the CA1/CA2 subfield of the dorsal hippocampus is essential for long-term spatial memory. J Neurosci. 1999, 19(9), 3535-3544).
In the Introduction, I suggest to take into account only animal framework, without referring to humans.
In the Introduction, the Authors should better explain the inhibitory avoidance task, reporting a more classical literature.
The paragraph “Psychological stress during adolescence affects enduring cognitive deficits in rodent [37]. Any alterations to adolescent social experiences may result in neurobehavioral measurements relevant to substance abuse, anxiety or depression [38]. It has been reported that early life stress in rodents or humans represents a neurodevelopmental risk with implications for subsequent cognitive abilities in adulthood [39]” is generic and out of focus. Why did the Authors refer to stress?
In the Methods\Results: As rightly reported, the performance in the inhibitory avoidance task is dependent on the amygdala and the hippocampus activity. However, it has been noted that the amygdala nucleus essential to allow the association between conditioned stimulus (dark compartment) and unconditioned stimulus (electric foot-shock) is the basolateral amygdala. Why did the Authors detect AR-immunoreactivity in the medial amygdala? At least, the Authors should explain how the basolateral amygdala (necessary for the inhibitory avoidance task) communicates with the medial amygdala (in which the ARs are densely expressed). Furthermore, hippocampus activity necessary for the inhibitory avoidance task requires more time. Why did the Authors measure the latency to reach the dark box only thirty minutes after the training, and not, for example, the day after? This would render the findings as more reliable and the hippocampus as the right brain structure in which search some effects. I suggest also to take into account the hypothalamus as hub in which found the effects of androgens.
The paragraph “2.2. Surgical procedure and steroidal manipulation” must be re-write. In fact, surgical procedures (sham and orchiectomy) must be reported. The Authors reported that the animals were purchased at 28 pnd and the day after their arrivals at the Facility, the animals were submitted to surgical procedures. It is not a good practice, given that may have influenced the results. Furthermore, what about the post-surgery recovery?
How can the Authors collect the trunk blood (presumably from the decapitated rats) prior to perfusion (methodology that requires the animals are not decapitated)?
In the paragraph 2.5. Western blotting, before the homogenization the amygdala or hippocampal areas have to be extracted. Please, always specify the methodologies in each step. Also, in the paragraph 2.6. Immunohistochemistry specific amygdala and hippocampus coordinates (according to a specific Rat Brain Atlas) lack.
In the Results, the Authors should show firstly the behavioral results, and then results of post-mortem experiments.
The One-way ANOVAs should be reported as F and P for the main effect, and only after the various P of post-hoc comparisons can be reported.
The Author very often reported decreased latency period in the illuminated box after electrical foot-shock. I retain this sentence as incorrected. In fact, as reported in the methods, the Authors measured the latency to reach the dark compartment, previously paired with the aversive unconditioned stimulus. If the Author desire to use the term illuminated box, they could report time spent in the illuminated box (before reaching the dark box).
More importantly, the serum androgen concentrations from OCX-T adolescent rats and from OCX-DHT rats were higher than sham-operated rats. Why?
In the paragraph 3.4. “Effects of AR-antagonist on serum steroid concentration, AR-expression and inhibitory 275 avoidance memory” the Authors reported “In western blotting (Fig. 3C-D), in comparison to intact adolescent rats (AR/a tubulin ratio; amygdala, 67.30 ± 4.18%; hippocampus, 63.64 ± 4.18%), the OCX rats showed a significant reduction in AR-expression both in amygdala (AR/a tubulin ratio; 24.95 ± 5.05%; ... Similarly, in immunohistochemistry, the both amygdaloid (Fig. 3E, G) and CA1 pyramidal neurons (Fig. 3F, H) of flutamide-treated rats exhibited a dramatic reduction in AR-immunoreactivity compared with intact adolescent rats”. What is the comparison? Flutamide-treated rats vs. vehicle-treated rats or vehicle- treated rats vs. OCX rats?
More interestingly, the serum androgen concentrations from OCX adolescent were lower than sham-operated rats, conversely the serum androgen concentrations from flutamide-treated rats were higher than vehicle-treated rats. How can the Author explain this difference?
I would also underline that, for all animals, in each group the latency to reach the dark compartment in the inhibitory avoidance task increased between training and test. This observation demonstrates that the orchiectomy did not impede the learning. Could be the behavioral results more related to anxiety than learning? I suggest to refer to learning more often than memory, because of the short times in which the Authors decided to perform the test after the training.
Minor comments.
In the Abstract, the Author cannot report any information regarding the basal excitability of amygdaloid/CA1 neurons, in fact they did not conduct electrophysiological experiments aimed at analyzing the cellular excitability.
Along the manuscript, the Authors should constantly maintain the acronyms (i.g. AR), explained the first time.
Reviewer 2 Report
- In the result section, most of the subtitles were written like "Inhibitory avoidance memory in sham-operated and steroid-manipulated rats". It is better to write as "Effect of inhibitory avoidance memory in rats". There is no need to mention sham-operated and steroid-manipulated rats.....etc
- Figure 1A, a-Tubulin leveling to be fixed.
- A schematic diagram of the study is recommended
Reviewer 3 Report
The work claims to identify new aspects of the influence of androgens on memory mechanisms. However, no new steps have been taken for this.
The short-term memory test was applied because the time interval between the presentation of the shock and the test was 30 minutes. The authors did not perform the test at other time intervals, for example, 24 hours after training, which would provide additional information about the effect of androgens on long-term memory. In addition, the work identified receptors in the hippocampus and amygdala, but there would be new data if other brain structures were examined. For these reasons, nothing new about memory mechanisms follows from the results of the work.